# Antimicrobial Peptide Pro10-1D Exhibits Anti-Allergic Activity: A Promising Therapeutic Candidate

**DOI:** 10.3390/ijms252212138

**Published:** 2024-11-12

**Authors:** Min Yeong Choi, Min Geun Jo, Keun Young Min, Byeongkwon Kim, Yangmee Kim, Wahn Soo Choi

**Affiliations:** 1Department of Immunology, College of Medicine, Konkuk University, Chungju 27478, Republic of Korea; 2Department of Bioscience and Biotechnology, Konkuk University, Seoul 05029, Republic of Korea; 3Institute of Biomedical Sciences & Technology, Konkuk University, Seoul 05029, Republic of Korea

**Keywords:** antimicrobial peptide, Pro10-1D, mast cell, allergy, passive cutaneous anaphylaxis, Src family kinase

## Abstract

Although antimicrobial peptides (AMPs) exhibit a range of biological functions, reports on AMPs with therapeutic effects in allergic disorders are limited. In this study, we investigated the anti-allergic effects of Pro10-1D, a 10-meric AMP derived from insect defensin protaetiamycine. Our findings demonstrate that Pro10-1D effectively inhibits antigen-induced degranulation of mast cells (MCs) with IC_50_ values of approximately 11.6 μM for RBL-2H3 cells and 2.7 μM for bone marrow-derived MCs. Furthermore, Pro10-1D suppressed the secretion of cytokines with IC_50_ values of approximately 2.8 μM for IL-4 and approximately 8.6 μM for TNF-α. Mechanistically, Pro10-1D inhibited the Syk-LAT-PLCγ1 signaling pathway in MCs and decreased the activation of mitogen-activated protein kinases (MAPKs). Pro10-1D demonstrated a dose-dependent reduction in IgE-mediated passive cutaneous anaphylaxis in mice with an ED_50_ value of approximately 7.6 mg/kg. Further investigation revealed that Pro10-1D significantly reduced the activity of key kinases Fyn and Lyn, which are critical in the initial phase of the FcεRI-mediated signaling pathway, with IC_50_ values of approximately 22.6 μM for Fyn and approximately 1.5 μM for Lyn. Collectively, these findings suggest that Pro10-1D represents a novel therapeutic candidate for the treatment of IgE-mediated allergic disorders by targeting the Lyn/Fyn Src family kinases in MCs.

## 1. Introduction

The prevalence of allergic diseases continues to increase, with 30–40% of the global population suffering from one or more allergic diseases [1,2]. Allergy refers to a condition where the immune system exhibits hypersensitivity to external substances such as pollen, bee venom, food, and drugs. Type 1 hypersensitivity encompasses disorders like anaphylaxis, allergic rhinitis, allergic asthma, food allergy, and atopic dermatitis. Among these, anaphylaxis presents severe symptoms that can be life-threatening [3].

Mast cells (MCs) are well recognized as key effector cells in Type 1 hypersensitivity. MCs are tissue-resident granulocytes located in close proximity to blood vessels, neurons, and lymphatic vessels, playing a role as sentinel cells for defense against invading antigens [4]. MCs express various types of activating and inhibitory receptors. In allergic reactions, MC activation can occur through two main pathways. In IgE-dependent allergic reactions, MCs are initially sensitized by IgE secreted from B cells activated by specific allergens. Upon subsequent exposure to the same allergen, activated MCs release pre-stored granule mediators such as histamine, tumor necrosis factor (TNF)-α, and proteoglycans, as well as de novo-synthesized lipid mediators. On the other hand, in IgE-independent allergic reactions, MCs can be activated through receptors such as Toll-like receptors (TLRs), MRGPRX2, and complement receptors, leading to the secretion of various inflammatory mediators including cytokines [5].

The binding of antigen to the IgE bound FcεRIα subunit on MCs initiates receptor aggregation, triggering intracellular signal transduction [6]. IgE-antigen crosslinking on the MC surface activates Src family kinases Lyn and Fyn. Specifically, activated Lyn phosphorylates tyrosine residues on the FcεRIγ subunit. The phosphorylated immunoreceptor tyrosine-based motif (ITAM) on FcεRIγ acts as a docking site for spleen-associated tyrosine kinase (Syk)’s two SH2 domains, inducing a conformational change that exposes Syk’s COOH-terminal region and increases its enzymatic activity [7]. Syk is phosphorylated by upstream tyrosine kinases Lyn/Fyn Src family kinases and undergoes further activation through self-autophosphorylation. Once sufficiently activated, Syk phosphorylates the linker for the activation of T cells (LATs), thereby functioning as a scaffold to form a multimolecular signaling complex including growth factor receptor-bound protein 2 (Grb2), lymphocyte cytosolic protein 2 (SLP-76), and phospholipase Cγ1 (PLCγ1) [8]. Activated PLCγ1, upon binding to LAT, hydrolyzes phosphatidylinositol 4,5-bisphosphate (PIP2) in the plasma membrane to produce diacylglycerol (DAG) and inositol 1,4,5-triphosphate (IP_3_). DAG activates protein kinase C (PKC), while IP_3_ increases intracellular Ca^2+^ concentration by releasing Ca^2+^ from endoplasmic reticulum stores through IP_3_ receptor binding. Subsequently, MCs undergo degranulation through these processes. Intracellular signals involving the activation of numerous members such as the PKC family and phospholipase D activate small GTPases like Rac, Ras, and Rho, which in turn induce the activation of extracellular signal-regulated kinase (Erk), c-Jun N-terminal kinase (JNK), and p38 mitogen-activated protein kinases (MAPKs). The activation of these MAPKs leads to the activation of various transcription factors, ultimately inducing the expression of inflammatory cytokines including interleukin (IL)-4, IL-5, and IL-13 [9].

Antimicrobial peptides (AMPs) are small peptides found in nature, typically composed of 10–60 amino acids. These peptides serve as the host’s first line of defense mechanism and possess immunomodulatory activities [10]. AMPs are secreted from cells of various species, including mammals, amphibians, microorganisms, and insects [11]. Due to their versatile characteristics, ongoing research focuses on developing AMPs as novel therapeutic agents that could potentially replace modern chemical drugs [12]. Defensins and cecropins stand out as prominent AMPs found in insects, constituting a crucial part of their innate immune system [13,14]. Protaetiamycine, a 43-amino acid defensin derived from the beetle *Protaetia brevitarsis*, is a cationic arginine-rich peptide known for its potent antibacterial activity [15].

Given the appeal of short antimicrobial peptides with potent antibacterial activity as potential therapeutic antibiotics, we developed the short 9-meric peptide Pro9-3, derived from protaetiamycine [16]. In this study, we utilized Pro10-1D, a 10-meric peptide developed previously by adding a D-arginine residue at the N terminal of Pro9-3D, an enantiomeric form of Pro9-3 [17]. Previous studies have shown that Pro10-1D is non-toxic to mouse macrophage RAW264.7 cells and exhibits inhibitory effects on sepsis induced by *E. coli* K1, gram-negative bacteria [18].

In this study, we first discovered that Pro10-1D inhibits degranulation and cytokine release in MCs stimulated by antigens through the inhibition of the Lyn/Fyn–Syk axis. Furthermore, we found that Pro10-1D dose-dependently inhibits IgE-dependent passive cutaneous anaphylaxis in mice. In conclusion, we propose that Pro10-1D holds promise as a novel peptide drug for the treatment of Type 1 hypersensitivity diseases.

## 2. Results

### 2.1. Effect of Pro10-1D on Antigen-Induced MC Degranulation

MCs, which belong to the myeloid lineage, are granulocytes characterized by the presence of numerous intracellular granules that contain histamine, serotonin, and a platelet-activating factor [19]. We first conducted experiments to investigate whether Pro10-1D inhibits MC degranulation induced by antigen (Ag) stimulation. Pro10-1D demonstrated a concentration-dependent inhibition of Ag-induced degranulation in both rat basophilic leukemia (RBL)-2H3 and bone marrow-derived mast cells (BMMC cells), with an inhibitory effect comparable to that of PP2, a well-established Src family kinase inhibitor, at a concentration of 10 μM (Figure 1A). The IC_50_ values were approximately 11.6 μM for RBL-2H3 cells and approximately 2.7 μM for BMMCs. To assess the reversibility of Pro10-1D’s inhibitory effect on MC degranulation, RBL-2H3 cells were incubated with Pro10-1D at 10 μM for 30 min, followed by three washing steps before Ag stimulation. The results revealed that the inhibitory effect of Pro10-1D was lost in the washed group, leading to the restoration of Ag stimulation-induced degranulation (Figure 1B). Importantly, Pro10-1D did not exhibit any signs of cellular toxicity, even at the highest experimental concentration (Figure 1C). These results suggest that Pro10-1D reversibly inhibits Ag-induced MC activation without compromising cell viability.

### 2.2. Effect of Pro10-1D on Activation of IgE-Mediated Signaling Pathway in MCs

We further investigated the effect of Pro10-1D on the inhibition of MC activation induced by IgE–Ag stimulation by examining its impact on the FcεRI-mediated signaling pathway. In RBL-2H3 cells, Pro10-1D exhibited concentration-dependent inhibition of Syk–LAT–PLCγ1 phosphorylation within the IgE-mediated signaling pathway activated by Ag stimulation (Figure 2A, left panel). Akt and MAPKs—Erk1/2, JNK, and p38—play crucial roles in the synthesis and secretion of inflammatory cytokines in MCs [20]. Additionally, Pro10-1D inhibited the antigen-stimulated phosphorylation of Erk1/2, JNK, and p38 in RBL-2H3 cells (Figure 2A, right panel). Notably, at concentrations of 5 μM or higher, Pro10-1D nearly completely blocked the Syk–LAT and Syk-mediated downstream signaling pathways in RBL-2H3 cells, compared to the control group treated with Ag alone. We extended this investigation to BMMCs to assess whether Pro10-1D exerts similar inhibitory effects. The results revealed that Pro10-1D inhibited the Syk and Syk-mediated downstream signaling pathways in BMMCs, mirroring the findings in RBL-2H3 cells (Figure 2B). Collectively, these findings demonstrate that Pro10-1D suppresses the activation of FcεRI-mediated signaling molecules in MCs following Ag stimulation.

### 2.3. Effect of Pro10-1D on the Release of Inflammatory Cytokines in BMMCs

Following IgE-mediated exocytosis of MC granules, late-phase responses occur several hours later and are characterized by the release of inflammatory cytokines such as tumor necrosis factor (TNF)-α and interleukin (IL)-4. TNF-α is a crucial cytokine responsible for the recruitment of leukocytes, including neutrophils and eosinophils, to inflammatory sites, thereby contributing to the progression of late-phase allergic reactions [21]. IL-4, a prototypical Th2 cytokine, is known to play a pivotal role in allergic responses by inducing leukotriene synthesis, MC proliferation, and survival [21,22]. We investigated whether Pro10-1D affects the release of inflammatory cytokines in Ag-stimulated BMMCs. The results demonstrated that Pro10-1D inhibited the release of the inflammatory cytokines IL-4 and TNF-α from Ag-activated BMMCs in a concentration-dependent manner. Specifically, Pro10-1D had IC_50_ values of approximately 8.6 μM for TNF-α (Figure 3A) and approximately 2.8 μM for IL-4 (Figure 3B). These findings suggest that Pro10-1D has the potential to modulate late-phase allergic reactions by suppressing the release of inflammatory cytokines associated with BMMC activation.

### 2.4. Effect of Pro10-1D in an IgE-Mediated Allergy Animal Model

To assess the regulatory effect of Pro10-1D on MCs in vivo, we employed the IgE-dependent passive cutaneous anaphylaxis (PCA) mouse model, a well-established model for Type I hypersensitivity immune reactions. IgE sensitization was carried out on the ears of mice for 12 h. Before the antigen challenge, the mice received an intraperitoneal injection of Pro10-1D for 1 h. Subsequently, the mice were intravenously injected with dinitrophenyl-human serum albumin (DNP-HSA) containing Evans blue dye. After the antigen challenge, dye extravasation was observed in the ear tissues of IgE-sensitized mice. The Evans blue dye was extracted from the ear tissues for analysis. The results revealed a dose-dependent reduction in dye extravasation in the Pro10-1D-treated group compared to the Ag-only group (Figure 4A,B) with an ED_50_ value of 7.6 mg/kg. These effects were similar to those of cetirizine, an antihistamine drug used clinically to inhibit Type I hypersensitivity symptoms (Figure 4A). Histological analysis was performed using ear tissue from PCA-induced mice. MC degranulation was assessed via toluidine blue O staining. The proportion of MCs undergoing degranulation upon Ag challenge in ear tissue demonstrated a dose-dependent decrease with Pro10-1D injection (Figure 4C). Overall, these findings indicate that Pro10-1D effectively inhibits allergic responses in an MC-dependent allergic disease mouse model.

### 2.5. Mechanism of Pro10-1D on IgE-Mediated Activation of MCs

Intracellular Ca^2+^ mobilization plays a critical role as a second messenger in the Lyn/Fyn/Syk axis-dependent signaling, which is essential for triggering MC activation and degranulation [23]. Ionomycin, a calcium ionophore, enhances Ca^2+^ influx upon MC stimulation [24]. Thapsigargin, a sarcoplasmic/endoplasmic reticulum calcium-dependent ATPase inhibitor, increases cytosolic calcium levels [25]. To investigate whether Pro10-1D affects the Ca^2+^ signal during MC activation, we stimulated BMMCs with ionomycin and thapsigargin and assessed changes in MC degranulation. However, Pro10-1D did not inhibit ionomycin- or thapsigargin-induced MC degranulation (Figure 5A). Additionally, PP2 exhibited a much weaker inhibitory effect on thapsigargin or ionomycin-induced degranulation compared to its inhibitory effect on antigen-induced degranulation (Figure 1). This inhibitory effect of PP2 under these conditions can be attributed to its known inhibitory activity against various other kinases in addition to Src family kinases [26]. Given our previous findings (Figure 2), we hypothesized that Pro10-1D may directly target upstream tyrosine kinases associated with the IgE receptor, thereby suppressing MC activation. To test this hypothesis, we proceeded to conduct in vitro kinase activity analysis focusing on Lyn and Fyn tyrosine kinases, which are key players in the FcεRI-mediated pathway triggered by Ag stimulation. The results demonstrated that Pro10-1D inhibited the activity of both Lyn and Fyn tyrosine kinases, with IC_50_ values of approximately 22.6 μM for Fyn and approximately 1.5 μM for Lyn (Figure 5B,C). These findings provide novel evidence that Pro10-1D directly inhibits the Lyn/Fyn tyrosine kinase axis, leading to the suppression of MC-dependent allergic responses.

## 3. Discussion

Previously, we developed Pro10-1D from protaetiamycine, an AMP derived from insects [18]. Pro10-1D was shown to exhibit potent antibacterial activity against multidrug-resistant (MDR) gram-negative bacteria by neutralizing lipopolysaccharide (LPS), thereby inhibiting inflammation in LPS-stimulated TLR4 signaling pathways. Furthermore, Pro10-1D demonstrated a restorative effect on multiple-organ damage and systemic inflammation in a sepsis mouse model [18]. Given the diverse functions of AMPs, including immune response regulation, we conducted experiments to explore whether Pro10-1D could also inhibit allergic responses. Our experimental findings, presented here for the first time, demonstrate that Pro10-1D inhibits Ag-induced activation of MCs.

Allergies are hypersensitive reactions triggered when foreign molecules, such as pollen, bee venom, food, and medications, enter our bodies. These abnormal immune responses, collectively referred to as hypersensitivity, are classified into four types: Immunoglobulin E (IgE)-mediated hypersensitivity (Type 1), IgG/IgM-mediated cytotoxic hypersensitivity (Type 2), immune complex-mediated hypersensitivity (Type 3), and cell-mediated hypersensitivity (Type 4) [27,28,29]. Meanwhile, in recent years, the prevalence of IgE-associated allergic disorders has risen globally. Current treatment options for allergic diseases include corticosteroids, antihistamines, and epinephrine [30]. Corticosteroids are among the most commonly used treatments and can be prescribed in various forms, such as nasal, oral, and topical cream. However, they are associated with side effects; nasal corticosteroids can cause irritation in the nose and throat, while oral corticosteroids may lead to side effects of anxiety and depression. Antihistamines, including H1 and H2 blockers, are also frequently prescribed in clinics depending on the target histamine receptor. However, these drugs carry the risk of side effects such as drowsiness and dry mouth, and overdoses of sedative antihistamines can induce seizures and arrhythmias. Omalizumab, an anti-IgE monoclonal antibody approved by the U.S. Food and Drug Administration (FDA), is utilized for asthma treatment, but its application in anaphylaxis management is limited [31]. While numerous anti-allergic drugs are available, research continues to focus on developing drugs with reduced side effects.

AMPs are small, naturally occurring antimicrobial peptides. In addition to their antimicrobial function, AMPs also play a role in regulating the host’s innate immune system. AMPs typically exert effects that inhibit the growth or induce the death of bacteria, fungi, and parasites [10]. Unlike conventional antibiotics, most AMPs employ a distinct mode of action for their antibacterial activity, targeting the pathogen’s membrane. These unique mechanisms make AMPs promising candidates for the development of next-generation antibiotics [11]. By August 2020, the antimicrobial peptide database cataloged a total of 3240 AMPs, including several approved by the FDA as antimicrobial drugs, such as bacitracin and dalbavancin. In addition, numerous natural and synthetic AMP derivatives are undergoing preclinical and clinical evaluations [12]. One of the advantages of AMP development as an antimicrobial drug lies in the ability to create new synthetic derivative drugs based on the structure-activity relationships of the AMP. However, research on AMP-based therapies for allergies has been relatively limited. Notably, human β-defensin-2(hBD2) has shown promise as a novel therapeutic agent for allergic asthma [32]. The intranasal application of hBD2 has been observed to reduce inflammatory cell influx into bronchoalveolar lavage fluid. An investigation into the immune regulatory action of AMPs revealed challenges related to the structural stability of AMPs within the body.

To address this, we have developed AMPs with increased resistance to various forms of hydrolysis, thereby enhancing their efficacy and stability in biological systems [16]. Pro10-1D, the AMP utilized in this study, is a modified all-D amino acid peptide, designed to enhance its stability. Our previous research demonstrated Pro10-1D’s immunoregulatory and antibacterial effects in sepsis induced by MDR gram-negative bacteria [18]. Building upon these findings, we investigated whether Pro10-1D could regulate the symptoms in the PCA mouse model, representing a Type I allergy response. The results showed that Pro10-1D reduced the PCA response in mice (Figure 4), indicating, for the first time, the potential of Pro10-1D as a novel AMP analog with anti-allergic activity.

Upon exposure to external substances, IgE-sensitized MCs are activated upon repeated encounters with the same substance. FcεRI-mediated MC activation is pivotal in initiating allergic reactions, setting off a series of cell signaling cascades that lead to calcium influx, degranulation, and the production of novel mediators [33]. Our study highlights the role and mechanism of Pro10-1D in inhibiting FcεRI-mediated MC signaling in RBL-2H3 cells and BMMCs. We discovered that Pro10-1D inhibits the Syk–LAT signaling pathway, which is an early signaling event in FcεRI cross-linking (Figure 2). Moreover, Pro10-1D reduced the phosphorylation of MAPKs, which are crucial for optimal MC degranulation and cytokine synthesis following Ag stimulation (Figure 2A). Building on these findings, we further examined whether Pro10-1D inhibits degranulation by modulating calcium flux—a downstream signaling pathway associated with the Syk/LAT/PLCγ axis in MCs. As a result, Pro10-1D did not inhibit MC degranulation triggered by thapsigargin or ionomycin (Figure 5A). This indicates that Pro10-1D regulates MC activation by inhibiting tyrosine kinases Lyn, Fyn, and Syk positioned adjacent to the IgE receptor, without affecting the Ca^2+^-dependent signaling pathway.

AMPs are recognized for their antibacterial activities as well as their immunomodulatory activities [34]. The mechanism for their immunomodulatory activity includes the regulation of inflammatory or anti-inflammatory cytokines and the induction of chemoattractants. For instance, papiliocin, an insect cecropin recognized as an anti-endotoxin AMP, alleviates the impact of *E. coli*-induced sepsis and exhibits immunomodulatory effects by inhibiting the TLR4-NFκB signaling pathway through competitive interference with the LPS-TLR4/MD-2 interaction [35]. LL-37, produced by neutrophils, neutralizes the pro-inflammatory activity of TLR ligands and modulates intracellular pathways, including the TLR-NFκB pathway while also demonstrating an anti-endotoxin effect. In addition, LL-37 induces IL-10 expression in T cells, B cells, and pDC cells [36,37]. Human neutrophil peptide1-3 has an immunomodulatory effect that promotes the secretion of TNF-α and IFN-γ from macrophages [38]. In this study, we have identified that Pro10-1D plays a critical role in inhibiting the activation pathway of Syk and Syk-dependent signaling molecules by Ag stimulation in MCs. Moreover, Pro10-1D demonstrated significant efficacy in suppressing the activity of Src family kinases, including Lyn and Fyn, which function as upstream signaling molecules to Syk, as evidenced through an in vitro kinase assay with IC_50_ values of approximately 1.5 μM for Lyn and 22.6 μM for Fyn (Figure 5B,C). Consequently, our findings propose a novel mechanism, by which Pro10-1D mitigates allergic responses through the inhibition of Src family kinases in MCs.

MCs play a pivotal role in the late-phase reaction following early-phase activation triggered by Ag stimulation [39]. During the late-phase response, MCs produce and secrete a range of cytokines and chemokines, which further intensify the inflammatory cascade initiated in the early response, leading to the recruitment and activation of various immune cells [40]. The late-phase response of MC activation is instrumental in sustaining and exacerbating allergic symptoms, such as bronchoconstriction and edema [41]. Suppressing MC-mediated late-phase responses, therefore, can help prevent the progression from acute allergic reactions to chronic inflammatory conditions. Consequently, inhibiting the secretion of MC cytokines offers promising avenues for managing allergic diseases and inflammatory disorders. MC-deficient mice engrafted with TNF-α^−/−^ BMMCs exhibited markedly reduced severity of ovalbumin (OVA)-induced airway hyperresponsiveness (AHR) compared to mice engrafted with wild-type (WT) BMMCs [42]. In this study, we found that Pro10-1D inhibits the release of pro-inflammatory cytokines, particularly IL-4 and TNF-α, which are indicative of the late-phase MC activation response (Figure 3A,B). Furthermore, MAPK signaling is a crucial pathway involved in the expression of inflammatory cytokines in MCs [43,44]. Targeting MAPKs in MCs, therefore, represents a potential therapeutic approach for regulating MC activation. For instance, roxatidine, the active metabolite of roxatidine acetate hydrochloride, has been shown to attenuate the release of pro-inflammatory cytokines by inhibiting the phosphorylation of NF-κB and p38 MAPK in MCs during IgE-independent stimulation [45]. Similarly, Pro10-1D reduced the phosphorylation of MAPKs—including p38, Erk, and JNK—following Ag stimulation, thereby attenuating the release of an inflammatory cytokine (Figure 2A and Figure 3). These findings also underscore Pro10-1D’s potential as a promising therapeutic candidate in the management of the late phase of allergic diseases.

To target intracellular mechanisms, therapeutic proteins must successfully traverse the plasma membrane. This can often be facilitated by using cell-penetrating peptides (CPPs), which typically consist of 5 to 30 amino acids [46,47]. The delivery capabilities of well-known CPPs such as TAT, R8, and Penetratin are well established. At physiological pH, these CPPs are predominantly positively charged, owing to the presence of multiple arginine and/or lysine residues. Arginine-rich cell-penetrating peptides are notably efficient in crossing the membranes of eukaryotic cells. For instance, R8 is made up of eight consecutive arginine residues, while the TAT peptide (sequence: YGRKKRRQRRR) contains six arginine and two lysine residues, contributing to its net positive charge—a critical factor for cellular uptake [46,47]. Pro10-1D has a sequence of rrlwlaiwrr-NH2. While we have not yet investigated the ability of Pro10-1D to enter mammalian cells, its configuration, including two arginines at each terminus, suggests a potential for cellular penetration. Furthermore, our in vitro kinase assay demonstrated that Lyn/Fyn kinase activity was inhibited (Figure 5B,C). Considering these results and the known cell membrane permeability of arginine-rich cell-penetrating peptides, it is plausible that Pro10-1D can penetrate the cell membrane and inhibit Lyn/Fyn kinase. However, the potential impact of Pro10-1D on the function of extracellularly expressed IgE receptors remains to be investigated.

## 4. Materials and Methods

### 4.1. Antibodies and Reagents

The chemical 4-amino-5-(4-chlorophenyl)-7-(dimethylethyl)pyrazolo[3,4-d]pyrimidine (PP2) was obtained from Calbiochem (La Jolla, CA, USA). Monoclonal dinitrophenol-specific IgE, DNP-human serum albumin (HSA), Evans blue, toluidine blue O, cetirizine, ionomycin, and thapsigargin were sourced from Sigma-Aldrich (St. Louis, MO, USA). Antibodies against the phosphorylated forms of Syk (Tyr352, 1:1000, #2717), LAT (Tyr220, 1:1000, #3584), PLCγ1 (Tyr783, 1:1000, #14008), Erk1/2 (Thr202/Tyr204, 1:1000, #9106), JNK (Thr183/Tyr185, 1:1000, #9251), p38 (Thr180/Tyr182, 1:1000, #9211) and total form of LAT (1:1000, #45533), Erk1/2 (1:1000, #9102), JNK (1:1000, #9252), p38 (1:1000, #9212), and β-actin: HRP-conjugated (1:5000, #5125) were purchased from Cell Signaling Technology (Danvers, MA, USA). Antibodies against Syk (1:200, #sc-51703) and PLCγ1 (1:200, #sc-7290) were obtained from Santa Cruz Biotechnology (Dallas, TX, USA). The medium used for cell culture was acquired from Welgene (Gyeongsan-si, Gyeongsangbuk-do, Republic of Korea).

### 4.2. Peptide Synthesis

Peptide synthesis was conducted as previously described [18]. Specifically, Pro10-1D (rrlwlaiwrr-NH_2_, where lowercase letters in the sequence represent D-amino acids) was synthesized using the N-(9-fluorenyl) methoxycarbonyl solid-phase synthesis method. The resulting peptide was purified to achieve over 95% purity through high-performance liquid chromatography (HPLC) utilizing a C18 column. During the synthesis and purification process, the peptide was obtained in its trifluoroacetate (TFA) salt form. Subsequently, a counter-ion exchange was conducted to convert the TFA salt to the acetate form of Pro10-1D. The molecular mass of Pro10-1D was determined via matrix-assisted laser desorption/ionization time-of-flight (MALDI-TOF) mass spectrometry, conducted on the Axima instrument (Shimadzu Scientific Instruments, Kyoto, Japan).

### 4.3. Animals

A cohort of male Balb/c mice, aged 5 weeks, was sourced from Orient Bio (Seongnam, Gyeonggi-do, Republic of Korea) for this study. The research was approved by the Institutional Animal Care and Use Committee (IACUC) of Konkuk University (permit number KU22156). All animal experiments were conducted in accordance with the National Institute of Health’s Guide for the Care and Use of Laboratory Animals.

### 4.4. Cultures of Rat Basophilic Leukemia (RBL)-2H3 Cells and Bone Marrow-Derived Mast Cells (BMMCs)

RBL-2H3 cells were obtained from the American Type Culture Collection (Manassas, VA, USA) and cultured in a complete minimal essential medium (MEM) supplemented with L-glutamine (100 Units/mL), penicillin (100 μg/mL), streptomycin (100 μg/mL), and 15% fetal bovine serum at 37 °C in a CO_2_-enriched environment. For the bone marrow-derived mast cell (BMMC) culture, bone marrow cells were harvested from the tibias and femurs of 5-week-old mice. These cells were then cultured in a complete RPMI 1640 medium, which contained 4 mM L-glutamine, 100 Units/mL penicillin, 100 μg/mL streptomycin, 1 mM sodium pyruvate, 0.1 mM non-essential amino acids, 25 mM HEPES, 50 mM 2-mercaptoethanol, 10% FBS, and 10 ng/mL IL-3. After 4 to 5 weeks of culture, the purity of BMMCs was assessed using flow cytometric analysis, focusing on FcεRIα and c-Kit expression as markers of mast cell maturation. Cells that were positive for both FcεRIα and c-Kit, constituting over 95% of the population, were identified as mast cells.

### 4.5. β-Hexosaminidase Assay

RBL-2H3 cells (2.0 × 10^5^ cells/well) were sensitized overnight with 20 ng/mL DNP-specific IgE in the culture medium. BMMCs (3.0 × 10^5^ cells/well) were sensitized with 100 ng/mL DNP-specific IgE for 4 h in the culture medium. Following sensitization, the cells were transferred to a Siraganian buffer (pH 7.2) for RBL-2H3 cells, containing 119 mM NaCl, 5 mM KCl, 40 mM NaOH, 25 mM PIPES, 5.6 mM glucose, 4 mM MgCl_2_, 1 mM CaCl_2_, and 0.5% BSA, or a Tyrode buffer (pH 7.4) for BMMCs, containing 135 mM NaCl, 5 mM KCl, 20 mM HEPES, 5.6 mM glucose, 1.8 mM CaCl_2_, 1 mM MgCl_2_, and 0.5% BSA. Pro10-1D was prepared in each buffer solution. After a 30 min incubation with or without Pro10-1D, the cells were stimulated with 25 ng/mL (RBL-2H3 cells) of DNP-human serum albumin (HSA) or 50 ng/mL (BMMCs) of DNP-HSA for 15 min at 37 °C. For the induction of MC activation using ionomycin or thapsigargin, BMMCs (3.0 × 10^5^ cells/well) were pre-incubated with Pro10-1D for 30 min without IgE sensitization. Subsequently, the cells were stimulated with 1 μM ionomycin or 300 nM thapsigargin for 15 min. A total of 30 μL of supernatant and 30 μL of p-nitrophenyl N-acetyl-β-D-glucosamine (p-NAG) were mixed with a 96-well plate and incubated at 37 °C for 70 min. The reaction was stopped by the addition of 250 μL of a 0.1 M carbonate buffer, and absorbance was measured at 405 nm. The degranulation values were calculated as the ratio of β-hexosaminidase activity secreted into the supernatant to the total β-hexosaminidase activity from both the supernatant and cell extract.

### 4.6. Cell Viability Assay

RBL-2H3 cells (2.0 × 10^4^ cells/well) were seeded in 96-well plates and incubated for 12 h, after which they were treated with Pro10-1D for 4 h. In parallel, BMMCs (2.0 × 10^4^ cells/well) were seeded in 96-well plates and incubated with Pro10-1D for 4 h. Cell viability was assessed using the Cell Counting Kit-8 (CCK-8) (Dojindo Laboratories, Kumamoto, Japan), following the manufacturer’s protocol. Absorbance was measured at 450 nm to determine cell viability.

### 4.7. Western Blotting

RBL-2H3 cells (8.0 × 10^5^ cells/well) were sensitized overnight with 20 ng/mL of DNP-specific IgE in a culture medium. Similarly, BMMCs (2.0 × 10^6^ cells/well) were sensitized for 4 h with 100 ng/mL of DNP-specific IgE. Following sensitization, the cell types were thoroughly washed with a fresh culture medium and treated with Pro10-1D for 30 min. Subsequently, the cells were stimulated with 25 ng/mL (for RBL-2H3 cells) or 50 ng/mL (for BMMCs) DNP-HSA for 15 min at 37 °C, after which the reaction was promptly terminated by placing the sample on ice for 5 min. RBL-2H3 cells were lysed in 70 μL of lysis buffer (pH 7.5), containing 20 mM HEPES, 150 mM NaCl, 1% Nonidet P–40, 10% glycerol, 60 mM octyl β-glucoside, 10 mM NaF, 1 mM Na_3_VO_4_, 1 mM phenylmethylsulfonyl fluoride, 2.5 mM nitrophenyl phosphate, 0.7 mg/mL pepstatin, and a protease inhibitor cocktail tablet. BMMCs were lysed using a RIPA buffer (Thermo Fisher Scientific, Waltham, MA, USA) supplemented with 1 mM PMSF, 0.7 μg/mL pepstatin, and a protease inhibitor cocktail (Sigma-Aldrich, St. Louis, MO, USA). The resulting cell lysates were subjected to high-speed centrifugation at 13,000 rpm for 10 min, after which the supernatant was mixed with a 4× Nu-PAGE^TM^ sample buffer (Thermo Fisher Scientific, Waltham, MA, USA) and denatured by heating at 96 °C for 5 min in a heating block. Protein separation was carried out using sodium dodecyl sulfate-polyacrylamide gel electrophoresis, followed by transfer to polyvinylidene fluoride membranes using transfer cassettes (Bio-Rad, Hercules, CA, USA). The membranes were subsequently incubated with specific primary antibodies in Tris-buffered saline (containing 0.1% Tween 20) supplemented with 5% BSA or skim milk. After primary antibody incubation, the membranes were incubated with horseradish peroxidase-labeled secondary antibodies. Protein bands were visualized using an enhanced Luminata Crescendo Western HRP substrate (EMD Millipore Corp., Billerica, MA, USA). The protein bands were identified horizontally, with a molecular weight marker serving as a reference to determine their positions.

### 4.8. Enzyme-Linked Immunosorbent Assay (ELISA)

BMMCs (2.0 × 10^6^ cells/well) were sensitized for 4 h with 100 ng/mL of DNP-specific IgE in a culture medium. Following sanitization, the cells were treated with Pro10-1D or PP2 for 30 min in the same culture medium. Subsequently, the cells were stimulated with 50 ng/mL DNP-HSA for 4 h at 37 °C. The concentrations of TNF-α and IL-4 in the culture medium were measured using Mouse ELISA Kits (Invitrogen, Waltham, MA, USA) according to the manufacturer’s instructions.

### 4.9. Passive Cutaneous Anaphylaxis (PCA)

Male Balb/c mice, aged approximately 6 weeks, were intraperitoneally injected using a 100 μL mixture (4:1 ratio) of alfaxan (alfaxalone 10 mg/mL; Jurox Inc., Rutherford, NSW, Australia) and rompun (xylazine hydrochloride 23.32 mg/mL; Bayer, Leverkusen, Germany) for general anesthesia. The mice then received intradermal injections of 50 ng of DNP-specific IgE, suspended in 10 μL of phosphate-buffered saline (PBS), into the ear skin. After 12 h, the mice were given intraperitoneally at 5, 10, and 20 mg/kg of Pro10-1D (suspended in PBS) based on prior preliminary experiments. For comparison, cetirizine was administered intraperitoneally at 20 mg/kg in mice. One hour after Pro10-1D administration, mice were intravenously injected with 100 μg of DNP-HSA suspended in 250 μL of a solution containing 5 mg/mL Evans blue. After 1 h of antigen challenge, the mice were euthanized humanely, and their ear tissues were excised. The extracted dye from these tissues was quantified by measuring its absorbance at 620 nm following incubation in 800 μL of formamide at 63 °C for 24 h.

### 4.10. Histologic Analysis

Following the PCA reaction, the ear skin of the mice was fixed using a solution of 4% paraformaldehyde in PBS. After fixation, the tissues were dehydrated in ethanol and embedded in paraffin. Serial sections, each 5 µm in thickness, were obtained from paraffin-embedded tissues and stained with toluidine blue. Representative images were captured at a total magnification of 200× to assess the percentage of degranulated MCs within the ear tissues. The total number of MCs and their degranulation were evaluated across 5–7 randomly selected sites from 5 to 7 sections per ear tissue (n = 5 mice). The results were expressed as the ratio of degranulated MCs to the total MC count within the tissue samples.

### 4.11. In Vitro Kinase Activity Assay

A radiometric Lyn, Fyn kinase assay was conducted at Eurofins (Celle-Lévescault, France) using KinaseProfiler^TM^. The assays were carried out at the Km concentration for ATP in dose response for each kinase. Briefly, the Lyn (mouse) assay was incubated using a mixture of 50 mM Tris pH 7.5, 0.1 mM EGTA, 0.1 mM Na_3_VO_4_, 0.1% β-Mercaptoethanol, 0.1 mg/mL Poly (Glu, Tyr) 4:1, 10 mM Magnesium acetate, and [γ^33^P]-ATP. The Fyn (human) assay was incubated with a mixture of 50 mM Tris pH 7.5, 0.1 mM EGTA, 0.1 mM Na_3_VO_4_, 250 µM KVEKIGEGTYGVVYK (Cdc2 peptide), 10 mM Magnesium acetate, and [γ^33^P]-ATP. The activity of Lyn and Fyn was initiated by the addition of the Mg/ATP mix, and the reactions were incubated for 40 min at room temperature. The reaction was halted by adding 0.5% phosphoric acid. A total of 10 μL of the reaction mixture was subsequently spotted onto a Filtermat A and washed four times for 4 min with 0.42% phosphoric acid, followed by washing in methanol. After drying, scintillation counting was performed.

### 4.12. Statistical Analysis

The results are presented as mean ± standard error of the mean (SEM), derived from three independent experiments. For the in vivo studies, a minimum of five mice were used per experimental group, while the in vitro experiments were performed in triplicate for each trial. Statistical analysis was performed using a one-way analysis of variance (ANOVA), followed by a Student’s *t*-test for unpaired data. Statistical significance was set at * *p* < 0.05 or ** *p* < 0.01 and all statistical computations were performed using GraphPad Prism 7 software (GraphPad Software Inc., San Diego, CA, USA). The IC_50_ and ED_50_ values were calculated using the AAT Bioquest calculator (accessed on 15 July 2023: https://www.aatbio.com/tools/ic50-calculator, https://www.aatbio.com/tools/ed50-calculator).

## 5. Conclusions

The findings from this study strongly support the effectiveness of the proposed peptide Pro10-1D in eliciting anti-allergic effects. By demonstrating its capacity to inhibit IgE-dependent MC activation and subsequent degranulation, as well as the release of pro-inflammatory cytokines such as IL-4 and TNF-α, Pro10-1D emerges as a potent candidate for therapeutic intervention in Type 1 hypersensitivity diseases. The mechanism by which Pro10-1D attenuates allergic responses through the inhibition of the Lyn/Fyn-Syk signaling axis further emphasizes its potential as a peptide drug in managing allergic reactions. Thus, Pro10-1D not only shows promise as an effective anti-allergic agent but also proposes a strategy for developing new therapies to address the growing prevalence of allergic diseases.

## Figures and Tables

**Figure 1 ijms-25-12138-f001:**
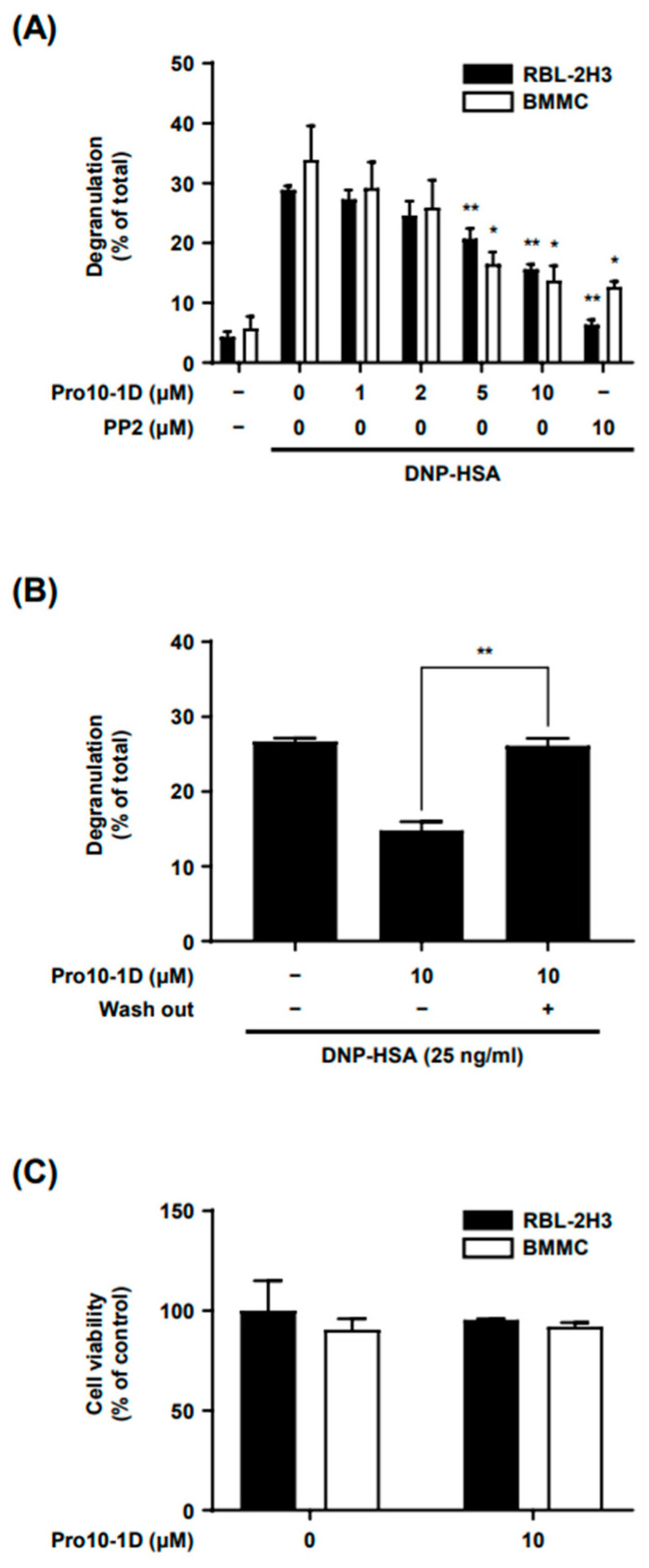
Pro10-1D inhibits IgE–Ag-induced degranulation in MCs. (**A**) RBL-2H3 cells and BMMCs were first sensitized with DNP-specific IgE and then treated with Pro10-1D or PP2 for 30 min, followed by the stimulation with DNP-HSA for 15 min. (**B**) IgE-sensitized RBL-2H3 cells were incubated with or without Pro10-1D for 30 min and then washed three times, followed by the stimulation with the antigen for 15 min. (**C**) The effects of Pro10-1D on cell viability were evaluated by incubating both RBL-2H3 cells and BMMCs with Pro10-1D for 4 h and measuring the absorbance at 450 nm using the CCK-8 kit. (**A**,**B**) The extent of degranulation from the MCs was determined by measuring the ratio of β-hexosaminidase activity released outside the cells to the total β-hexosaminidase activity. The data are presented as the mean ± SEM from three independent experiments, each performed in triplicate. Statistical significance is indicated by an asterisk; * *p* < 0.05, ** *p* < 0.01. PP2 is a general Src family kinase inhibitor.

**Figure 2 ijms-25-12138-f002:**
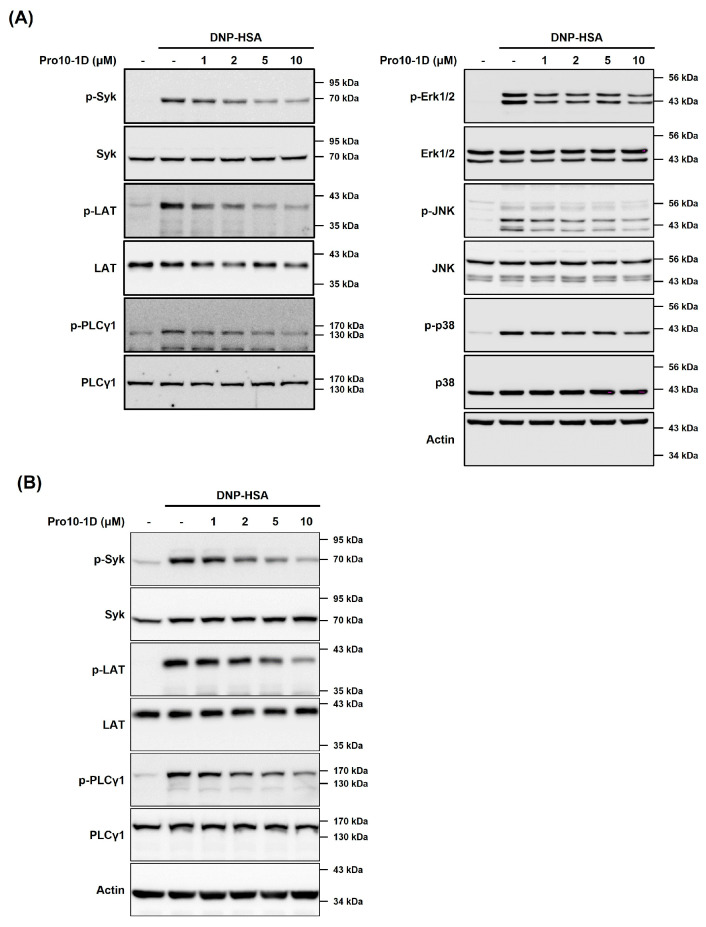
Pro10-1D suppresses the activation of FcεRI-mediated signaling proteins in MCs. (**A**) RBL-2H3 cells (8.0 × 10^5^ cells/well) or (**B**) BMMCs (2.0 × 10^6^ cells/well) were sensitized with DNP-specific IgE and then treated with Pro10-1D for 30 min, followed by the stimulation with 25 ng/mL and 50 ng/mL DNP-HSA, respectively, for 15 min. Western blotting was performed as described in the Section 4 “Materials and Methods”. The images shown are representative results from three independent experiments.

**Figure 3 ijms-25-12138-f003:**
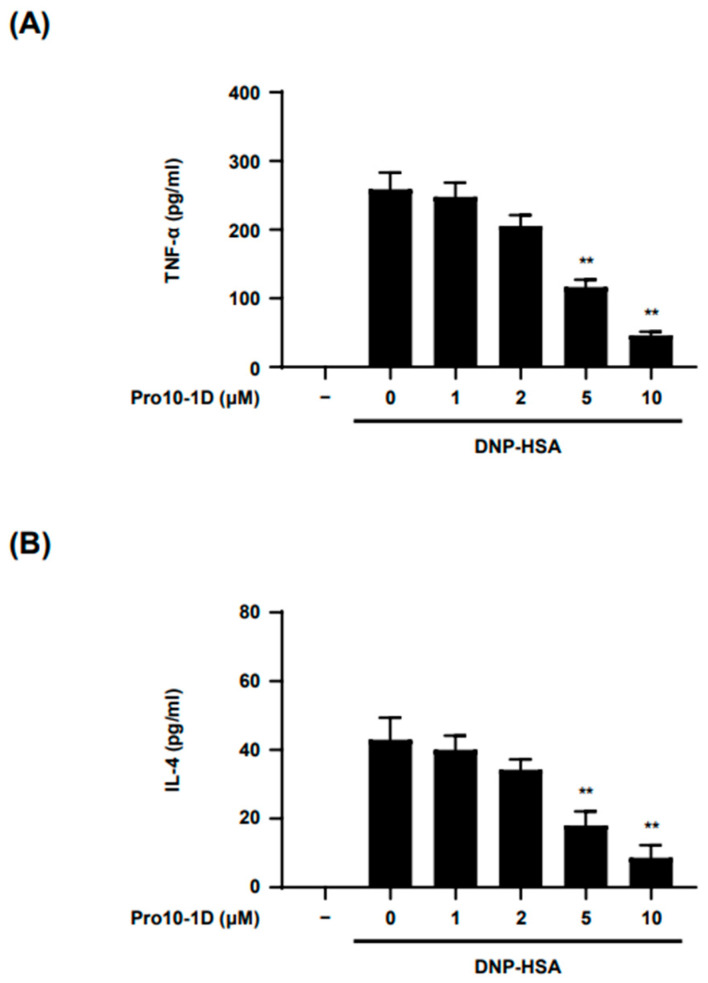
Pro10-1D decreases the release of inflammatory cytokines in BMMCs by antigen stimulation. (**A**,**B**) BMMCs (2.0 × 10^6^ cells/well) sensitized with DNP-specific IgE were treated with Pro10-1D for 30 min and then stimulated with 50 ng/mL DNP-HSA for 3 h. After incubation, the concentrations of IL-4 and TNF-α released into the culture media were subsequently measured by ELISA. The data are represented as the mean ± SEM from three independent experiments. Statistical significance is indicated by an asterisk; ** *p* < 0.01.

**Figure 4 ijms-25-12138-f004:**
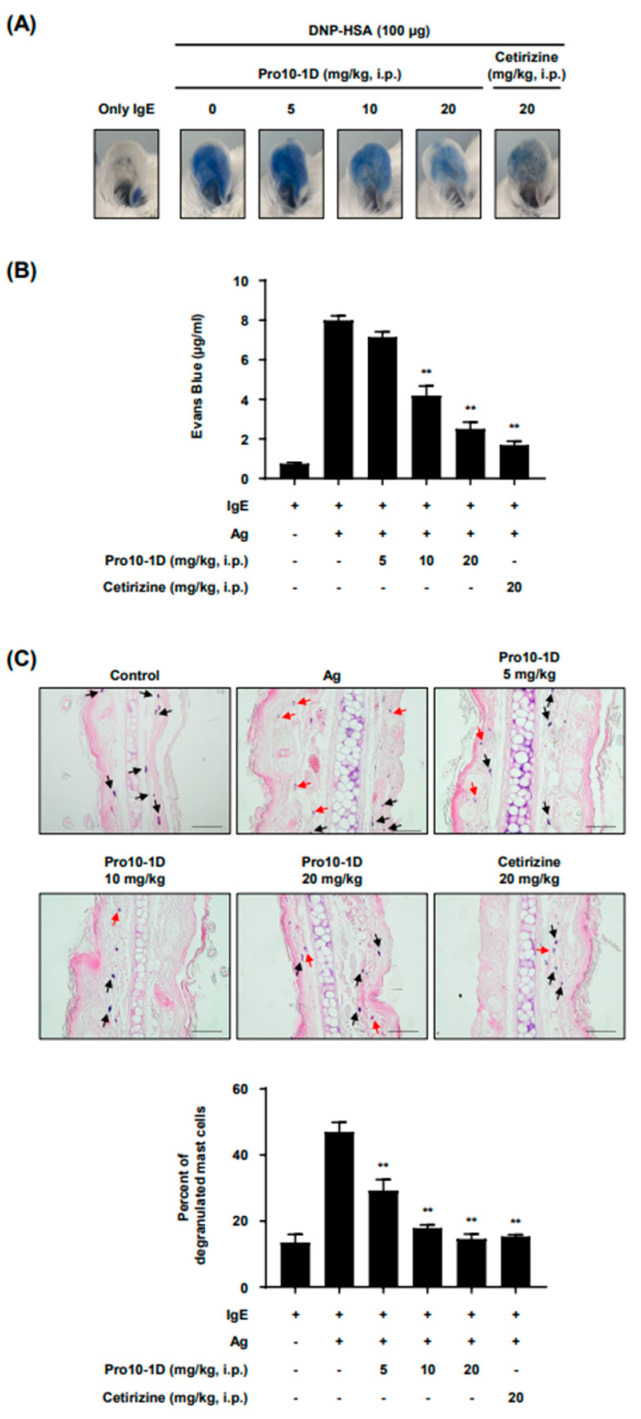
Pro10-1D diminishes the IgE-mediated passive cutaneous anaphylaxis (PCA) reaction. Mice were intradermally injected with DNP-specific IgE (50 ng) in their ear. After 12 h, they were intraperitoneally injected with Pro10-1D at concentrations of 5, 10, and 20 mg/kg, with cetirizine (20 mg/kg) serving as a reference drug. One hour later, DNP-HSA with Evans blue was administered by intravenous injection. The images in (**A**) depict representative ear images for each group. (**B**) Evans blue dye extravasation was quantified by absorption at 620 nm. (**C**) The ears of the mice from the experiment depicted in (**A**) were processed for histological analysis. Ear sections were stained with toluidine blue-eosin, and degranulated MCs were counted. The scale bar in histological images represents 25 μm. The red arrows indicate degranulated MCs and the black arrows indicate intact MCs. The representative images were captured at a total magnification of 400×. The representative images (**A**,**C**) and values (**B**,**C**) were generated from three independent experiments. The values (**B**,**C**) are presented as the mean ± SEM. Statistical significance is indicated by an asterisk; ** *p* < 0.01.

**Figure 5 ijms-25-12138-f005:**
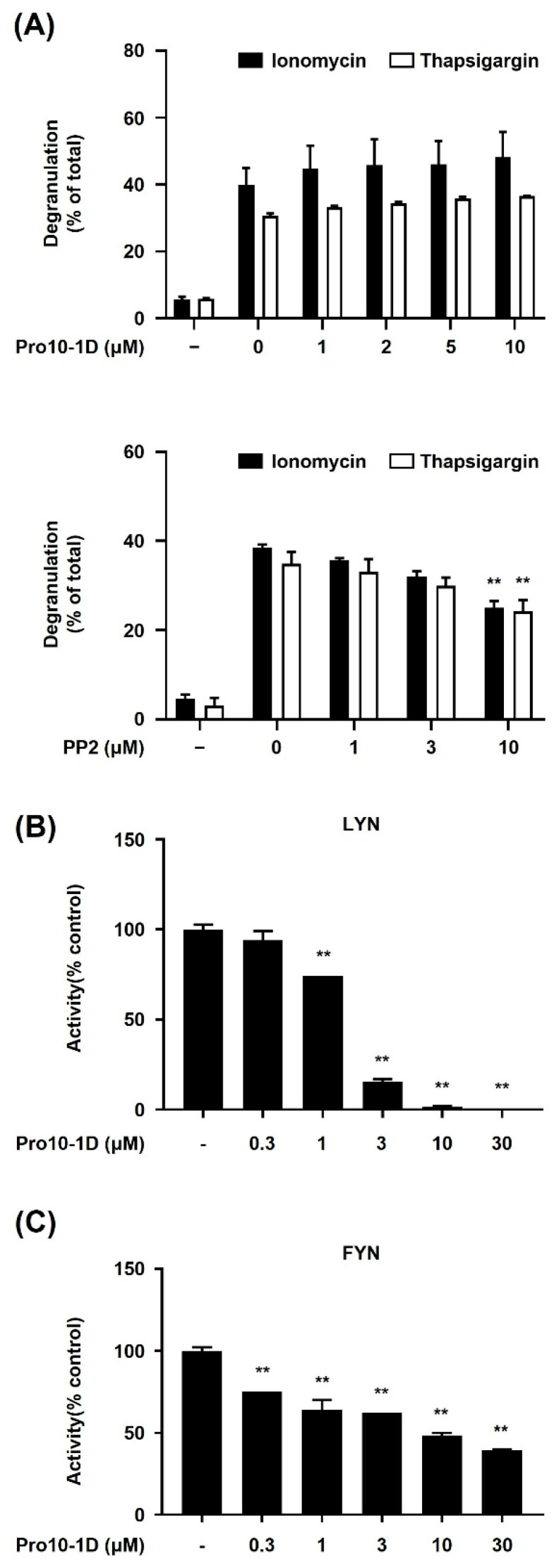
Pro10-1D suppresses the activity of Lyn and Fyn kinases in vitro. (**A**) BMMCs (3.0 × 10^5^ cells/well) were pre-incubated with Pro10-1D or PP2 for 30 min and then stimulated with 1 μM ionomycin or 300 nM thapsigargin for 15 min. (**B**,**C**) KinaseProfiler radiometric protein kinase assays were performed with Lyn and Fyn kinase. The protocols used for the kinase assay are detailed in the Section 4 “Materials and Methods”. (**A**–**C**) The data are presented as the mean ± SEM from two independent experiments. Statistical significance is indicated by an asterisk; ** *p* < 0.01.

## Data Availability

The original contributions presented in this study are included in the article; further inquiries can be directed to the corresponding author/s.

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
