# Peer review of "Antimicrobial Peptide Pro10-1D Exhibits Anti-Allergic Activity: A Promising Therapeutic Candidate"

_ijms, 2024, doi:10.3390/ijms252212138_

Round 1

Reviewer 1 Report

Comments and Suggestions for Authors

In this paper, the authors examined the effects of antimicrobial peptide Pro10-1D on mast cell activation, and found that Pro10-1D inhibited the antigen-induced mast cell activation probably via inhibiting Lyn and Fyn.

The findings were interesting but I have several concerns.

Major comments:

1) Pro10-1Δ is α cationic arginine-rich peptide. If so, can it enter the cells? It is important to clarify whether this peptide enters into the cells and inhibits Lyn/Fyn, or binds to DNP-HSA or IgE at outside of cells and inhibits the interaction of the antigen to IgE.

2) To comparison, the authors should show whether PP2 inhibits ionomycin- and thpasigargin-induced degranulation.

3) Although cetirizine can inhibit mast cell degranulation at high concentrations, it inhibits evans blue extravasation via antagonizing H1 receptor.  Was the data that the inhibition of mast cell degranulation at this dose reproducible?

Minor comments:

1) Section 2.4: The authors should comment about cetirizine.

Author Response

We thank the reviewer for their helpful and invaluable comments. In response, we have made appropriate revisions to address all concerns raised by the reviewer. The changes in the text are noted in red. Our responses (in red text) to the reviewers’ individual comments (in italics) are given below.

Comment #1: Pro10-1Δ is α cationic arginine-rich peptide. If so, can it enter the cells? It is important to clarify whether this peptide enters into the cells and inhibits Lyn/Fyn, or binds to DNP-HSA or IgE at outside of cells and inhibits the interaction of the antigen to IgE.

Response: We sincerely appreciate this valuable comment. To target intracellular mechanisms, therapeutic proteins must successfully traverse the plasma membrane. This can often be facilitated by the use of cell-penetrating peptides (CPPs), which typically consist of 5 to 30 amino acids [46,47]. The delivery capabilities of well-known CPPs such as TAT, R8, and Penetratin are well established. At physiological pH, these CPPs are predominantly positively charged, owing to the presence of multiple arginine and/or lysine residues. Arginine-rich cell-penetrating peptides are notably efficient in crossing the membranes of eukaryotic cells. For instance, R8 is made up of eight consecutive arginine residues, while the TAT peptide (sequence: YGRKKRRQRRR) contains six arginine and two lysine residues, contributing to its net positive charge—a critical factor for cellular uptake [46,47]. In contrast, Penetratin adopts an alpha-helical structure near the membrane, with several studies suggesting that the incorporation of its hydrophobic segments into the membrane is vital for its uptake [46].

Additionally, there are antimicrobial peptides characterized by high cationic properties and richness in arginine residues. Cationicity plays a significant role in their antibacterial activity by targeting negatively charged bacterial membranes. Pro10-1D has a sequence of rrlwlaiwrr-NH2. While we have not yet investigated the ability of Pro10-1D to enter mammalian cells, its configuration, including two arginines at each terminus, suggests a potential for cellular penetration. If valid, this could highlight the therapeutic potential of Pro10-1D. Due to time constraints for this revision, we regret that we could not explore this critical aspect. However, we plan to label the peptide by synthesizing FITC-Pro10-1D to evaluate its penetration capability in future studies, which will help determine whether it directly inhibits Lyn/Fyn or binds to DNP-HSA or IgE outside of cells. Moreover, our in vitro kinase assay demonstrated the suppression of Lyn/Fyn kinase activity. Given these findings and the established ability of arginine-rich cell-penetrating peptides to traverse cell membranes, we hypothesize that Pro10-1D is capable of penetrating the cell membrane and exerting an inhibitory effect on Lyn/Fyn kinase. Nevertheless, we cannot entirely exclude the possibility that Pro10-1D might have a partial impact on the function of cell surface-expressed IgE receptors.

We appreciate the reviewer's insights again and we have addressed this comment by mentioning the possible mechanisms on CPP (page 15, lines 325 - page 16, line 341).

Comment #2: To comparison, the authors should show whether PP2 inhibits ionomycin- and thpasigargin-induced degranulation.

Response: As per the reviewer's comment, we conducted additional experiments. The results showed that PP2 exhibited a much weaker inhibitory effect on thapsigargin or ionomycin-induced degranulation compared to its inhibitory effect on antigen-induced degranulation (Fig. 1). Although PP2 is a src family kinase inhibitor, it can also inhibit various other kinases within the cell, which may explain its partial inhibitory effect under these conditions (please see Biochem. J. 2003 371(Pt 1):199-204. doi: 10.1042/BJ20021535.). However, taken together, PP2 also appears to inhibit antigen-induced degranulation through a more specific inhibition of Lyn/Fyn Src family kinases, which are upstream kinases for the calcium signaling pathway, in mast cells. Comparing these results, we suggested that Pro10-1D also inhibits antigen-stimulated degranulation by inhibiting the upstream signals of the calcium signaling pathway. We have added these results to Figure 5A (see Fig. 5A and page 9, lines 187-191).  

Comment #3: Although cetirizine can inhibit mast cell degranulation at high concentrations, it inhibits evans blue extravasation via antagonizing H1 receptor. Was the data that the inhibition of mast cell degranulation at this dose reproducible? 

Response: We agree with the reviewer's comment. In our animal experiments, we used cetirizine, a clinically used anti-allergic drug, as a reference for the anti-allergic response. Our initial intention was to use cetirizine as an anti-histamine drug, not as a mast cell stabilizer, since anti-histamines are typically the first-line treatment in clinical settings. As the reviewer pointed out, it has been reported that cetirizine can inhibit mast cells at very high concentrations (approx. 1 mM). As suggested by the reviewer, we re-examined the inhibitory effect of cetirizine on antigen-induced mast cell degranulation, but we could not find any inhibitory effect at lower concentrations (below 100 μM). Therefore, theoretically, it is difficult to assume that cetirizine can act as a mast cell stabilizer in vivo. However, in repeated in vivo PCA-induced ear tissue histology experiments, we consistently observed that antigen-stimulated mast cell degranulation was inhibited by cetirizine administration (see Fig. 4C). However, we cannot currently fully explain the mechanism by which cetirizine inhibits mast cell degranulation in vivo. Therefore, we believe that further research on the detailed mechanism would be very interesting. We would like to thank the reviewer for providing such a valuable comment and giving us a good idea for further experiments. However, since this is not the main focus of this paper, we did not mention it to avoid distracting from the main points of the paper.

Comment #4: Section 2.4: The authors should comment about cetirizine. 

Response: As suggested by the reviewer, this is described in the main text (page 8, lines 171-172).

We believe that the comments from this reviewer were very reasonable and accurate. We have made revisions based on your comments, which significantly improved the quality of this paper. We deeply appreciate the thorough review by the reviewer who carefully read our manuscript.

Reviewer 2 Report

Comments and Suggestions for Authors

Dear Authors,

1. Authors must show the levels of anti-inflammatory cytokines (IFN-gamma, IL-10 etc) after treatment with peptide during allergic reactions as they have key role in suppressing pro-inflammatory cytokines during allergic response.

2. Please elaborate the conclusion section with emphasis on effectiveness of the propsed peptide in anti allergic effects.

Author Response

We thank the reviewer for their helpful and invaluable comments. In response, we have made appropriate revisions to address all concerns raised by the reviewers. The changes in the text are noted in red. Our responses (in red text) to the reviewers’ individual comments (in italics) are given below.

Comment #1: Authors must show the levels of anti-inflammatory cytokines (IFN-gamma, IL-10 etc) after treatment with peptide during allergic reactions as they have key role in suppressing pro-inflammatory cytokines during allergic response.

Response: We appreciate the reviewer’s insightful comments. Based on our findings, we hypothesize that Pro10-1D exerts its effects by inhibiting the activity of Lyn and Fyn, early signaling Src family kinases, in antigen-stimulated mast cells. Consequently, the inhibition of early signaling events in mast cells likely results in the suppression of degranulation and cytokine expression/secretion. Therefore, we anticipate that Pro10-1D would also inhibit the secretion of anti-inflammatory cytokines from mast cells. However, studies investigating the antigen-induced secretion of IL-10 and IFN-gamma in mast cells are limited. Although we attempted to examine the secretion of these cytokines in bone marrow-derived mast cells as suggested by the reviewer, we were unable to detect their production. Nevertheless, the reviewer’s comment highlights the intriguing possibility of investigating the long-term effects of repeated in vivo administration of Pro10-1D on the expression and secretion of the anti-allergic cytokines IFN-gamma and IL-10. We plan to conduct such studies in our future research. Thank you!

Comment #2: Please elaborate the conclusion section with emphasis on effectiveness of the proposed peptide in anti allergic effects. 

Response: Thank you for your insightful comment. We have elaborated on the conclusion section to highlight the effectiveness of the proposed peptide in anti-allergic effects in conclusion (page 16, lines 343-352).  

We believe that the comments from this reviewer were very reasonable and accurate. We have made revisions based on your comments, which significantly improved the quality of this paper. We deeply appreciate the thorough review by the reviewer who carefully read our manuscript.

Round 2

Reviewer 1 Report

Comments and Suggestions for Authors

I accept for the authors comments.